# AIMHI: Protecting Sensitive Data through Federated Co-Training

**Amr Abourayya**
Institute for AI in medicine (IKIM)
and Ruhr-University Bochum
University Hospital Essen
Essen, Germany
`amr.abourayya@uk-essen.de`

**Michael Kamp**
Institute for AI in medicine (IKIM)
and Ruhr-University Bochum and Monash University
University Hospital Essen
Essen, Germany
`michael.kamp@uk-essen.de`

**Jens Kleesiek**
Institute for AI in medicine (IKIM)
University Hospital Essen
Essen, Germany
`jens.kleesiek@uk-essen.de`

**Erman Ayday**
Case Western Reserve University
Cleveland, USA
`exa208@case.edu`

**Kanishka Rao**
Carenostics
Pennsylvania, USA
`kanishka.rao@carenostics.com`

**Geoff Webb**
Monash University
Melbourne, Australia
`michael.kamp@uk-essen.de`

**Bharat Rao**
Carenostics
Pennsylvania, USA
`bharat.rao@carenostics.com`

## Abstract

Federated learning offers collaborative training among distributed sites without sharing sensitive local information by sharing the sites' model parameters. It is possible, though, to make non-trivial inferences about sensitive local information from these model parameters. We propose a novel co-training technique called AIMHI that uses a public unlabeled dataset to exchange information between sites by sharing predictions on that dataset. This setting is particularly suitable to healthcare, where hospitals and clinics hold small labeled datasets with highly sensitive patient data and large national health databases contain large amounts of public patient data. We show that the proposed method reaches a model quality comparable to federated learning while maintaining privacy to high degree.

## 1 Introduction

Can we collaboratively train models from distributed sensitive datasets while maintaining data privacy at a level required in healthcare? Federated learning [12] allows distributed sites, e.g., hospitals or clinics, to collaboratively train a joint model without directly disclosing their sensitive data by instead periodically sharing model parameters. An attacker or curious observer can, however, make inferences about local data from model parameters [11] and model updates [21]. Differential privacy provides a rigorous and measurable privacy guarantee [5] that can be achieved by perturbing model

36th Conference on Neural Information Processing Systems (NeurIPS 2022).

parameters appropriately [20]. This perturbation, however, can reduce model quality, resulting in a trade-off between privacy and quality. As we show in our experiments, even with substantial perturbation one can infer membership of a training sample [17] with high probability, i.e., whether a data point is present in a local dataset from the model parameters shared in federated learning.

We propose to instead use a distributed co-training approach [9], where sites train local models and exchange predictions on a shared unlabeled dataset, instead of sharing model parameters. By forming a consensus from shared predictions, one obtains pseudo-labels for the shared unlabeled dataset that can be used for local training. Iterating this process improves the consensus, and thereby the quality of pseudo-labels, effectively nudging local models to come to an agreement. We show in our experiments that this approach, which we call AIMHI (AI Models for Healthcare Improvement), achieves the same model quality as federated learning (FedAvg [12]), but protects privacy to a high level—membership inference is significantly less likely compared to vanilla federated learning and federated learning with differential privacy. These results indicate that, given a public unlabeled dataset, this approach constitutes a more favorable trade-off between privacy and model quality.

The AIMHI approach requires a large shared unlabeled dataset, and is in spirit similar to distributed distillation [2]. While such unlabeled public datasets are not always available, in healthcare, large public health databases are quite common: the US NCHS databases, the UK's NHS databases, the UK Biobank [18], the MIMIC-III database [6], or the planned European EHDS contain vast amounts of patient data that can be used in many application scenarios.

Our contributions are

(i) a novel distributed co-training approach to collaboratively train models from privacy-sensitive distributed data sources, and

(ii) a preliminary empirical evaluation of model quality and privacy on the CIFAR10 benchmark dataset, indicating high model quality and a substantial improvement in privacy.

## 2 Preliminaries

We assume learning algorithms $\mathcal{A} : \mathcal{X} \times \mathcal{Y} \to \mathcal{H}$ that trains a model $h \in \mathcal{H}$ using a dataset $D \subset \mathcal{X} \times \mathcal{Y}$ from an input space $\mathcal{X}$ and output space $\mathcal{Y}$, i.e., $h = \mathcal{A}(D)$. Given a set of $m \in \mathbb{N}$ clients with local datasets $D^1, \ldots, D^m \subset \mathcal{X} \times \mathcal{Y}$ drawn iid from a data distribution $\mathcal{D}$ and a loss function $\ell : \mathcal{Y} \times \mathcal{Y} \to \mathbb{R}$, the goal is to find a single model $h^* \in \mathcal{H}$ that minimizes the risk

$$\mathcal{E}_{\mathcal{D}}(h) = \mathbb{E}_{(x,y) \sim \mathcal{D}}(\ell(h(x), y)) \ .$$

In centralized learning, the datasets are pooled as $D = \bigcup_{i \in [m]} D^i$ and $\mathcal{A}$ is applied to $D$, usually to minimize the empirical risk

$$\mathcal{E}_{emp}(h, D) = \sum_{(x,y) \in D} \ell(h(x), y) \ .$$

In federated learning (FL) we assume that the learning algorithm is iterative [cf. Chp. 2.1.4 7], i.e., $\mathcal{A} : \mathcal{X} \times \mathcal{Y} \times \mathcal{H} \to \mathcal{H}$ that updates a model $h_{t+1} = \mathcal{A}(D, h_t)$. In this case, centralized learning means applying $\mathcal{A}$ to $D$ until convergence. Note that applying $\mathcal{A}$ on $D$ can be the application to any random subset, e.g., as in mini-batch training, and convergence is measured in terms of low training loss, small gradient, or small deviation from previous iterate.

In standard federated learning [12], $\mathcal{A}$ is applied in parallel for $b \in \mathbb{N}$ rounds on each client locally to produce local models $h^1, \ldots, h^m$. These models are then centralized and aggregated using an aggregation operator $\mathfrak{a} : \mathcal{H}^m \to \mathcal{H}$, i.e., $\overline{h} = \mathfrak{a}(h^1, \ldots, h^m)$. The aggregated model $\overline{h}$ is then redistributed to local clients which perform another $b$ rounds of training using $\overline{h}$ as a starting point. This is iterated until convergence of $\overline{h}$ with the goal to minimize the empirical risk over all local datasets[12], i.e.,

$$\mathcal{E}_{emp}(h, D^1, \ldots, D^m) = \frac{1}{m} \sum_{k=1}^{m} \mathcal{E}_{emp}(h, D^k) = \frac{1}{m} \sum_{k=1}^{m} \sum_{(x,y) \in D^k} \ell(h(x), y) \ .$$

When aggregating by averaging, this method is also known as federated averaging. Next, we describe our proposed distributed co-training approach.

**Algorithm 1:** AIMHI

**Input:** communication period $b$, learning algorithm $\mathcal{A}$, $m$ clients with local datasets
       $D^1, \ldots, D^m$, unlabeled shared dataset $U$, total number of rounds $T$

**Output:** final models $h_T^1, \ldots, h_T^m$

1   initialize local models $h_0^1, \ldots, h_0^m$
2   $P \leftarrow \emptyset$
3   **Locally** *at client $i$ at time $t$* **do**
4       $h_t^i \leftarrow \mathcal{A}(D_i \cup P, h_{t-1}^i)$
5       **if** $t \% b = b - 1$ **then**
6            $L^i \leftarrow h_t^i(U)$
7            send $L^i$ to server
8            receive $L$ from server
9            $P \leftarrow (U, L)$
10      **end**
11  **At server** *at time $t$* **do**
12       receive local pseudo-labels $L^1, \ldots, L^m$
13       $L \leftarrow \text{consensus}(L^1, \ldots, L^m)$
14       send $L$ to all clients

## 3   AIMHI: Distributed Co-Training

We propose a semi-supervised, distributed co-training approach that collaboratively trains models via sharing predictions. It uses an unlabeled dataset $U$, producing pseudo-labels $L$ for it by forming a consensus of the predictions of all local models. Unlabeled data and pseudo-labels form an additional public, shared dataset $P$ that is combined with local data for training. The details are described in Alg. 1: at each client $i$, the local model is updated using the local dataset $D^i$ combined with the current pseudo-labeled public dataset $P$. The updated model is used to produce improved pseudo-labels $L^i$ for the unlabeled data $U$, which are sent to a server every $b$ rounds. At the server, as soon as all local prediction $L^1, \ldots, L^m$ are received, a consensus $L$ is formed and broadcasted back to the clients. Forming a consensus is similar to obtaining a prediction from an ensemble [4]. For our classification experiments, we use vanilla majority voting [3]. Note that more elaborate consensus mechanisms offer a rich design space for improvements. On receiving the new consensus labels $L$ from the server, the client updates the public pseudo-labeled dataset $P$ and performs another iteration of local training.

## 4   Empirical Evaluation

We perform a preliminary empirical evaluation of the performance of AIMHI in comparison to vanilla federated learning (i.e., FedAvg) and federated learning with differential privacy on the CIFAR10 image classification dataset [10]. For that, we measure their prediction accuracy on a test set, as well as their privacy vulnerability.

### 4.1   Privacy Vulnerability

We measure privacy vulnerability by performing membership inference attacks against AIMHI and FL according to two different attack scenarios per approach. In both attacks, the attacker creates an attack model using a model it constructs from its training and test datasets. Similar to previous work [17], we assume that the training data of the attacker has a similar distribution to the training data of the client. Once the attacker has its attack model, it uses this model for membership inference. In blackbox attacks (in which the attacker does not have access to intermediate model parameters), it only uses the classification scores it receives from the target model (i.e., client's model) for membership inference. On the other hand, in whitebox attacks (in which the attacker can observe the intermediate model parameters), it can use additional information in its attack model. Since the proposed AIMHI does not reveal intermediate model parameters to any party, it is only subject to blackbox attacks. Vanilla federated learning on the other hand is subject to whitebox attacks. Each inference attack produces a

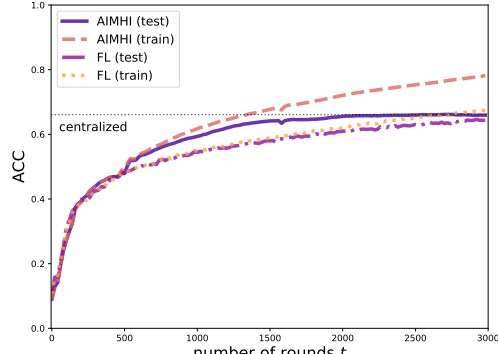

Figure 1: Test accuracy (ACC) over time on CIFAR10 with ACC of average model (FL) and average ACC of local models for AIMHI.

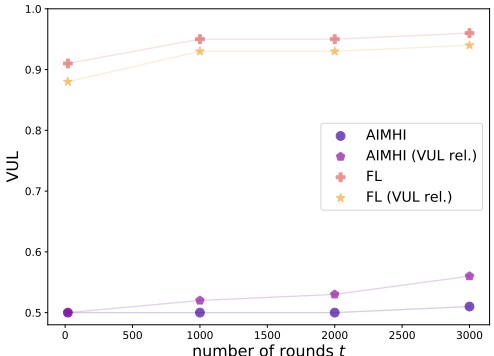

Figure 2: Privacy vulnerability (VUL) over time for AIMHI and FL for realistic and relaxed attack scenarios.

membership score of a queried data point, indicating the likelihood of the data point being a member of the training set. We measure the success of membership inference as ROC AUC of these scores. The **vulnerability (VUL)** of a method is the ROC AUC of membership attacks over $K$ runs over the entire training set (also called attack epochs) according to the attack model and scenario. A vulnerability of $1.0$ means that membership can be inferred with certainty, whereas $0.5$ means that deciding on membership is a random guess.

We assume the following attack model: clients are honest and the server may be semi-honest (follow the protocol execution correctly, but it may try to infer sensitive information about the clients). The main goal of a semi-honest server is to infer sensitive information about the local training data of the clients. This is a stronger attacker assumption compared to a semi-honest client since the server receives the most amount of information from the clients during the protocol, and a potential semi-honest client can only obtain indirect information about the other clients. We also assume that parties do not collude. The attack scenarios are[1]:

- *AIMHI realistic*: the attacker can send a (forged) unlabeled dataset to the clients and observe their predictions, equivalent to one attack epoch ($K = 1$);

- *FL realistic*: the attacker receives model parameters and can run an arbitrary number of attacks—we use $K = 500$ attack epochs;

- *AIMHI relaxed*: the attacker can send a (forged) unlabeled dataset in each communication round, albeit to different models in each round—we simulate this by assuming pessimistically that models are sufficiently similar over all rounds and set $K = T/b = 150$;

- *FL relaxed*: the attacker cannot copy model parameters on their machine, but can only perform an attack epoch in each communication round, thus being able to perform $K = T/b = 150$ attack epochs after $T = 3000$ rounds.

To measure vulnerability, we use the ML Privacy Meter tool [13]. This tool allows us to quantify the privacy risks associated with machine learning models by performing a range of membership inference attacks [14] on machine learning models and measuring attack success. It simulates different levels of access and model knowledge for attackers, e.g., limiting attackers to only predictions, or loss values, or assuming they have access to the model's parameters. For our experiments, we assume that for AIMHI the attacker only has access to the predictions (i.e., output of the last layer), while for FL, the attacker can access all layers.

## 4.2 Experimental Setup

We use the common CIFAR10 dataset [10] which consists of $50000$ training and $10000$ test images with 10 classes. We use $10000$ samples drawn iid from the training images as training set, $40000$ images as unlabeled dataset and the $10000$ test images as test set. We use $m = 5$ clients, each with

---

[1]Note that the relaxed scenarios are more in favor of FL than the realistic ones.

a local training set of $n = 2000$ samples. Clients use a convolutional neural network (details are provided in the Appendix) and communicate every $b = 20$ rounds for FL and AIMHI. Both FL and AIMHI are run for $T = 3000$ rounds.

The experiments are implemented in TensorFlow [1], the code is publicly available[2]. For our experiments, we use a simple CNN architecture given in Table 1. As optimizer we use Adam with learning rate $\alpha = 0.001$, exponential decay rates $\beta_1 = 0.9$ and $\beta_2 = 0.999$, and $\epsilon = 10^{-7}$.

| Layers | Activation function |
|---|---|
| Conv2D(32, 3, 3) | relu |
| MaxPooling2D(2, 2) | |
| Conv2D(64, 3, 3) | relu |
| MaxPooling2D((2, 2) | |
| Flatten layer | |
| Dense(64) | relu |
| Dense(10) | softmax |

Table 1: Model architecture for AIMHI and federated learning.

### 4.3 Differential Privacy for Federated Learning

A common defense against membership inference attacks is applying appropriate clipping and noise before sending models. This guarantees $\epsilon, \delta$-differential privacy for local data [20] at the cost of a slight-to-moderate loss in model quality. This technique is also proven to defend against backdoor and poisoning attacks [19]. We compare AIMHI against federated learning with differential privacy through the Gaussian mechanism (DP-FL). For that, each client $i$ first clips their parameters $w^i$ to

$$w_c^i = \frac{w^i}{\max\left\{1, \frac{\|w^i\|}{C}\right\}}$$

for a constant $C > 0$, and then add Gaussian noise $\tilde{w}^i = w_c^i + \epsilon$ with $\epsilon \sim \mathcal{N}(0, \sigma)$ for $\sigma \geq 0$. The level of privacy depends on the choice of $C$ and $\sigma$ [20].

### 4.4 Results

The results presented in Table 2 show that AIMHI and FL achieve comparable test accuracy after $T = 3000$ rounds[3]. Note that centralized training on the 10000 training samples achieves a test accuracy of $0.661$, so FL and AIMHI both achieve virtually optimal model quality. At the same time, FL is vulnerable to membership attacks, both in the realistic (VUL) and the relaxed (VUL(rel.)) attack scenarios with a vulnerability of over $0.9$. Note that since the attacker can run an arbitrary number of attacks, the privacy vulnerability is in principle $1.0$, we instead measure the practical vulnerability under a large, but limited number of attacks. AIMHI on the other hand preserves privacy ($VUL = 0.51$) in the realistic attack scenario, and still has very little vulnerability ($0.56$) under the relaxed scenario. Adding noise for differential privacy reduces vulnerability considerably to around $0.85$, but at the cost of accuracy which drops to around $0.42$. We investigate the convergence in terms of test accuracy in Figure 1: Both AIMHI and FL converge quickly, with AIMHI converging slightly faster. Looking at the development of privacy vulnerability over time, we observe that vulnerability for FL is already high after the first communication round (i.e., after $t = 20$ rounds), as shown in Figure 2, and increases to $0.94$ after $t = 1000$ rounds. This holds also for the relaxed scenario, although with slightly less vulnerability. For AIMHI realistic instead the vulnerability remains low

---

[2]https://anonymous.4open.science/r/AIM_HI

[3]For FL, we report the test accuracy of the average model; for AIMHI, we report the average of test accuracies for local models—we observe that at $T = 3000$ their variance is 0.

| | Accuracy | VUL | VUL (relaxed) |
|---|---|---|---|
| **AIMHI** | **0.659** | **0.51** | 0.56 |
| **FL** | 0.643 | 0.96 | 0.94 |
| **DP-FL**$(C = 2.0, \sigma = 0.01)$ | 0.425 | 0.85 | 0.87 |

Table 2: Test accuracy (ACC) and privacy vulnerability (VUL) for AIMHI and federated learning, both vanilla federated averaging (FL) and federated averaging with differential privacy (DP-FL) on $m = 5$ clients with local training set size $|D^1| = \cdots = |D^m| = 8\,000$ and an unlabeled dataset of size $|U| = 10\,000$.

(0.5 to 0.51). For AIMHI relaxed we observe an increase in vulnerability, as expected, since we assume in this scenario that the attacker performs a number of attacks proportional to the number of communication rounds.

## 5 Discussion and Conclusion

We presented a novel distributed co-training approach for privacy-preserving federated learning that protects sensitive local datasets, where vanilla federated learning is susceptible to privacy attacks, at the same time achieving similar model quality as FL. Our initial experiments support the hypothesis that distributed co-training can be competitive with FL, given a large unlabeled dataset, while preserving data privacy to a much higher degree.

Co-training requires a shared unlabeled dataset, which is not available in all application scenarios. In healthcare, however, it is not uncommon to have large quantities of unlabeled data points available. A limitation of AIMHI is that local datasets must be sufficiently large to create useful local models [cf. 8]—otherwise, the poor quality pseudo-labeled dataset will not improve local training. Choosing more elaborate consensus methods [15, 16] is an interesting direction for future work that can improve performance even with small local datasets. An important advantage of co-training is that local models do not have to have the same architecture, as FL requires. In fact, local models can be arbitrary. This includes interpretable models, like decision trees or rule ensembles, for which no FL method exists so far. Exploring distributed co-training for such model classes could open a whole new avenue for federated learning.

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
