# OpenReview forum: "AIMHI: Protecting Sensitive Data through Federated Co-Training"
_NeurIPS.cc/2022/Workshop/Federated_Learning — FL-NeurIPS 2022 Poster_

### Official Review · Reviewer_BhHe · 2022-10-16
**AIMHI: Algorithm Description not Clear**

The authors present AIMHI, a distributed co-training algorithm for Healthcare applications. The proposed model uses a joint dataset shared by each node that allows the authors to develop a training algorithm that avoids model sharing and hence might lead to privacy saving. The paper is difficult to understand in the current writing, specifically, it is not clear how the algorithm is implemented. Detailed comments are listed below:

- The description of the algorithm is not at all clear. In the introduction and the abstract, the authors mention that they will use a shared dataset for distributed co-training, but in Section 3 the authors suddenly introduce an additional unlabeled dataset L, the reason for introducing this additional dataset is not clear.
- The algorithm steps are not clear, like what does step 5: $t \\% b = b - 1$ mean. The authors present the local updates in an abstract manner without clarifying what those steps are. Also, the meaning of pseudo-labels in Section 3 and in the algorithm steps is not clear.
- In Lines 61-62 the authors mention that if the local models are averaged, the algorithm is referred to as Federated Averaging (FedAvg). This statement is not true, the algorithm is referred to as FedAvg only if the local models are updated using SGD. Please correct the statement.
- Please correct "citepwei2020federated" in Line 18.

Although the experiments show strong results, it is not totally clear what the authors are trying to accomplish.

---

### Official Review · Reviewer_JfNH · 2022-10-16

In the work, the authors propose a co-training technique relying on the consensus of predictions on a large public unlabeled dataset instead of shared model parameters, which helps protect data privacy and defend the membership inference attack. While the method requires the availability of a large-scale public unlabeled dataset, it is still a good fit in the healthcare scenario.

---

### Official Review · Reviewer_jK33 · 2022-10-18

This paper proposes a novel distributed co-training approach to jointly train models from privacy-sensitive data and can achieve high model quality with maintaining privacy.

The paper is well-structured and easy to follow. It contains the intuition and real-world feasibility of using unlabeled public datasets in cross-silo settings. The paper provides preliminary empirical evaluations for measuring privacy vulnerability.


Several comments:
1.  	It is not clear about the advantage (from the side of both algorithm design and experimental results) compared to the distillation methods in FL. It would be better to make those comparisons.
2.  	For the proposed AIMHI algorithm, the attack scenarios are on the unlabeled dataset, and then it assumes the attacker only has access to the (unlabeled dataset) predictions. However, if the attacks are on the local clients’ private dataset, it seems like the effectiveness of the proposed algorithm would be greatly reduced.
3.  	As the experimental setup uses the Adam optimizer, there are some adaptive federated learning papers that could be referred:
[1] Reddi, S., et al. (2020). Adaptive federated optimization.
[2] Tong, Q., Liang, G., and Bi, J. Effective federated adaptive gradient methods with non-iid decentralized data.
[3] Wang, Y., Lin, L., & Chen, J. (2022). Communication-Efficient Adaptive Federated Learning.

---

### Decision · Program_Chairs · 2022-10-20

Accept (Poster)